# Heteroaryl-Ethylenes as New Lead Compounds in the Fight against High Priority Bacterial Strains

**DOI:** 10.3390/antibiotics10091034

**Published:** 2021-08-25

**Authors:** Dafne Bongiorno, Nicolò Musso, Paolo G. Bonacci, Dalida A. Bivona, Mariacristina Massimino, Stefano Stracquadanio, Carmela Bonaccorso, Cosimo G. Fortuna, Stefania Stefani

**Affiliations:** 1Section of Microbiology, Department of Biomedical and Biotechnological Sciences (BIMETEC), University of Catania, 95123 Catania, Italy; nmusso@unict.it (N.M.); dalidabivona@gmail.com (D.A.B.); m.cri1503@gmail.com (M.M.); s.stracquadanio@hotmail.it (S.S.); stefanis@unict.it (S.S.); 2Department of Chemical Sciences, University of Catania, 95125 Catania, Italy; paolo.g.bonacci@gmail.com (P.G.B.); cg.fortuna@unict.it (C.G.F.)

**Keywords:** heteroaromatic stilbene derivatives, antimicrobial activity, *Staphylococcus aureus*, *Enterococcus faecalis*, *Escherichia coli*, *Pseudomonas aeruginosa*, *Acinetobacter baumannii*, *Klebsiella pneumoniae*

## Abstract

The widespread use of antibiotics has led to a gradual increase in drug-resistant bacterial infections, which severely weakens the clinical efficacy of antibacterial therapies. In recent decades, stilbenes aroused great interest because of their high bioavailability, as well as their manifold biological activity. Our research efforts are focused on synthetic heteroaromatic stilbene derivatives as they represent a potentially new type of antibiotic with a wide antibacterial spectrum. Herein, a preliminary molecular modeling study and a versatile synthetic scheme allowed us to define eight heteroaromatic stilbene derivatives with potential antimicrobial activity. In order to evaluate our compound’s activity spectrum and antibacterial ability, minimum inhibitory concentration (MIC) and minimum bactericidal concentration (MBC) tests have been performed on Gram-positive and Gram-negative ATCC strains. Compounds PB4, PB5, PB7, and PB8 showed the best values in terms of MIC and were also evaluated for MBC, which was found to be greater than MIC, confirming a bacteriostatic activity. For all compounds, we evaluated toxicity on colon-rectal adenocarcinoma cells tumor cells (CaCo2), once it was established that the whole selected set was more active than 5-Fluorouracil in reducing CaCo-2 cells viability. To the best of our knowledge, the biological assays have shown for these derivatives an excellent bacteriostatic activity, compared to similar molecular structures previously reported, thus paving the way for a new class of antibiotic compounds.

## 1. Introduction

The World Health Organization (WHO) has identified antimicrobial resistance (AMR) as a matter of concern for the public authorities in both developed and emerging countries. To guide the research and development of new and effective antibiotic treatments, the WHO has developed in 2017 a global priority pathogens list (global PPL) of antibiotic-resistant bacteria [1]. Amongst the Gram-positive species, *Staphylococcus aureus* (methicillin-resistant-MRSA, vancomycin intermediate- hVISA and VISA, and resistant-VRSA) and *Enterococcus* spp. (vancomycin-resistant or VRE) were classified as high priority. Amongst Gram-negative species, those of greatest interest are *Escherichia coli* resistant to cephalosporins, fluoroquinolones, and aminoglycosides; *Acinetobacter baumannii* resistant to carbapenem and fluoroquinolones; *Peudomonas aeruginosa* with combined resistance (resistance to piperacillin ± tazobactam, ceftazidime, fluoroquinolones, aminoglycosides, and carbapenems) and carbapenem resistant *Klebsiella pneumoniae* (KPC).

It is crucial to remember the importance of antibiotics research for modern medicine to counter the growing threat posed by AMR. With the continuous rise of difficult-to-treat pathogens and related infectious diseases, the development of new strategies [2,3,4] and improved effective therapeutics, including antibiotics and their combinations, are urgently needed [5,6,7,8].

The 2015–2019 report of the European Antimicrobial Resistance Surveillance Network (EARS-Net) [9] indicates that AMR is highly variable depending on bacterial species, antimicrobial group, and geographical region. Although a high level of resistance is still observed, variations were moderate for most Gram-negative bacterial species. Moreover, even if MRSA rates are stabilizing or decreasing in most European countries, *S. aureus* remains an important pathogen as it is one of the most common causes of bloodstream infections, with a high burden in terms of morbidity and mortality [10,11]. *Enterococcus* spp. are commensal in the human gastrointestinal tract. *E. faecium* and *E. faecalis* can cause a variety of infections, such as urinary tract infections, bloodstream infections, and endocarditis. They are intrinsically resistant to a broad range of antimicrobial agents (i.e., cephalosporins and sulphonamides) and showed low susceptibility to beta-lactams. They can acquire genes conferring high-level resistance to aminoglycosides. Finally, the increase of glycopeptide resistance is of increasing clinical relevance. Glycopeptide resistance is mediated by the VanA gene that conferred high level resistance to vancomycin and a variable level of resistance to teicoplanin, and VanB, that conferred a variable level of resistance to vancomycin [12]. *E. coli* is part of the normal intestinal microbiota and is a common cause of severe infections such as bloodstream and urinary tract infections; resistance to antibiotics is associated with mutations (fluoroquinolones) or mediated by the acquisitions of mobile genetic elements (production of extended spectrum beta-lactamases-ESBLs and carbapenemase) [13]. *Klebsiella pneumoniae* is mainly found in the human gastrointestinal tract, skin, and respiratory tract, and causes healthcare-associated infections. *K. pneumoniae* can acquire resistance genes through plasmids and, as *E. coli,* can have genes in their chromosome that encode class A beta-lactamase [14]. *P. aeruginosa* is a ubiquitous bacterium that can become an opportunistic pathogen in hospitalized and immunocompromised individuals. It commonly causes healthcare-associated pneumonia, bloodstream infections, and urinary tract infections. *P. aeruginosa* is intrinsically resistant to the majority of antimicrobial agents due to its selective ability to prevent various antibiotic molecules from penetrating its outer membrane or to extrude them if they enter the cell. The antimicrobial groups that remain active include fluoroquinolones, aminoglycosides, some beta-lactams (e.g., piperacillin-tazobactam, ceftazidime, cefepime, ceftolozane-tazobactam, ceftazidime-avibactam), and polymyxins. Resistance of *P. aeruginosa* to these agents can be acquired through one or more of several mechanisms, including modified antimicrobial targets, efflux and reduced permeability, and degrading enzymes [15]. Species belonging to the *Acinetobacter* spp. are opportunistic pathogens primarily associated with healthcare-associated infections. Species belonging to the complex are intrinsically resistant to most antimicrobial agents due to their selective ability to prevent various molecules from penetrating their outer membrane. The antimicrobial groups that remain active include some fluoroquinolones (e.g., ciprofloxacin), aminoglycosides (e.g., gentamicin), carbapenems, polymyxins, sulbactam, and tigecycline. Resistance to antibiotics can lead to mutation in the chromosome or plasmid acquisition [13,16,17].

Proceeding on this track, we aimed to define new antimicrobial candidates starting from an in-house database of heteroaryl-ethylenes which were previously tested in vitro for their antitumor activity on different human cancer cell lines [18,19,20]. The stilbene structure can be considered a privileged scaffold as it occurs frequently both in biologically active natural compounds (i.e., resveratrol) and in synthetic derivatives [21,22,23,24,25], for applications ranging from the visualization of biomolecules in living systems and real-time tracking of cellular events to fluorescent dye guidance during surgeries. These systems offer several advantages, including easy, rapid, efficient, and inexpensive synthetic procedures; good ADME properties; and high photo-stability.

Further exploration of these structures, from both a synthetic and a microbiological perspective, will be the focus of this work. A preliminary antimicrobial structure-activity relationship (SAR) investigation was performed, using forty stilbenic and heteroaromatic compounds whose antimicrobial activity towards *S. aureus* was previously reported [22,26,27,28,29,30,31,32,33,34,35], in order to select eight promising candidates (PB1-8, see Appendix A).

The spectrum of ability of the PBn compounds to inhibit the proliferation of bacterial cultures was evaluated through MIC (minimal inhibition concentration) and MBC (minimal bactericidal concentration) assays on a selected sample of Gram-positive and Gram-negative ATCC strains. We also tested scalar inoculum concentrations to assess if the molecules had inoculum effect. All compounds were tested for their cytotoxicity on colon-rectal adenocarcinoma tumor cells (CaCo2).

Our preliminary results are highly encouraging and pave the way for further investigations of heteroaryl-ethylenes as antimicrobial agents against difficult-to-treat Gram-positive and Gram-negative pathogens. These novel compounds may be both safe and effective in the treatment of several Gram-positive bacterial infections, especially those caused by MRSA, VRE, and *Acinetobacter baumannii*.

## 2. Results

### 2.1. VolSurf+ Analysis

The antibacterial effects of the compounds reported in Appendix A were evaluated in silico using the Volsurf+ software (see Appendix A for further details) [36,37,38,39,40,41]. The data for the microbiological activity, expressed as MIC (mg/L), of 40 heteroaryl and stilbenoids compounds (Appendix A) tested on *S. aureus* ATCC29213 constituted our library for the prediction of the antibacterial activity [22,26,27,28,29,30,31,32,33,34,35]. The partial least squares (PLS) model obtained allowed us to locate the most active compounds in the upper left side of the 2D scores plot (Figure 1). The biological/chemical space represented by the PLS t1/t2 scores plot was properly explored and the model showed high predictive accuracy (R^2^ = 0.88). The leave-one-out (LOO) procedure was used to validate the model internally and to evaluate its predictive ability; a Q^2^ value of 0.36 was obtained.

We used this model for the external prediction of more than seventy-five structures from our in-house database of heteroaryl-ethylenes (yellow circles in Figure 1 inset indicate the predicted compounds) [18,19,20]; we selected six promising candidates (PB1–6) [18,20] and designed two new molecules (PB7–8). This statistical and molecular modelling approach allowed us not only to highlight a promising scaffold (positioned in the lower left-hand corner of the 2D scores plot in Figure 1 but also to define key molecular properties, as molecular interaction fields (MIFs)-based descriptors that correlate with activity for drug candidates.

The percentage of unionized species (%FU), 3D pharmacophoric descriptors (DODODO, ACDODO, and DRDODO) and those related to H-bond donors volume (WO1 and WO2) are directly correlated with the observed antimicrobial activity. SKIN, ID1-2, and LogPc-Hex descriptors (referring to ADME properties such as skin permeability, to physico-chemical properties such as distance of the hydrophobic volume from the center of mass and to the partition coefficient water/cyclohexane) are inversely correlated with the measured biological activity. The compound showing the most promising activity is PB4; however, all structures of the PB series should have good bacteriostatic activity. These preliminary results guided the synthesis of the new compounds and the in vitro evaluation of the designed test set.

### 2.2. Synthesis of PBn Compounds

The strategy used to synthesize the new designed molecule, PB7–8, involves Knoevenagel condensation between the proper heteroaromatic aldehyde and 2-methyl-5,6,7,8-tetrahydroisoquinolin-2-ium iodide; see Appendix A for the detailed experimental procedure.

### 2.3. Antimicrobial Activity of PBn Compounds against Control Strains

The spectrum of ability of the PBn compounds to inhibit the proliferation of bacterial cultures was evaluated by means of standard MIC (minimal inhibition concentration) and MBC (minimal bactericidal concentration) assays.

To assess the activity of the selected compounds, we included in the study other control strains selected for their resistance to antibiotics or their ability to infect and damage the host, in order to have a sample as varied as possible.

#### 2.3.1. Antimicrobial Activity of PBn Compounds against Gram-Positive Strains

We included four *S. aureus*: ATCC12598–MSSA as the standard for the MIC evaluation of *S. aureus*; the Mu50 strain, as it was a VISA strain; USA300 as it was a community acquired MRSA; and ATCC12598, as it was a particularly invasive MSSA strain. For *Enterococcus* spp. we included *E. faecalis* ATCC29212–VSE as the standard for MIC evaluation and the ACTT51299 as it was an *E. faecalis* VRE.

As reported in Table 1, all strains showed good MIC values for compounds PB4, PB5, PB7, and PB8; to the best of our knowledge, these compounds showed the lowest values reported to date for stilbenoid compounds. None of the six strains tested against these compounds showed a difference in MIC of more than two dilution factors. The compounds that yielded the best MIC values (MIC ≤ 16 mg/L), namely PB4, PB5, PB7, and PB8, were tested for their bactericidal activity. The results of the MBC assays are also listed in Table 1.

Remarkable was the comparison between MIC and MBC values, e.g., PB4 always showed three values of difference; PB5 and PB7 had variable values for Staphylococci, while they have been shown to have a difference of 4 to 5 dilutions in Enterococci; PB8 showed no significant differences between MICs and MBCs in either case. 

Considering the situation where MIC and MBC values coincide, we could elaborate that the action of the substance in question was bactericidal, whereas if there was a difference, the action of the substance in MIC values was only bacteriostatic. Translating this concept to the present work, a comparison can be made for each individual strain by looking at which compound had the lowest MBC value at the same MIC: an explicative example was shown with *E. faecalis* ATCC29212 and *E. faecalis* ATCC51299, in which PB5 and PB8 have the same MIC, but PB8 does not induce a bacteriostatic effect only, but also a bactericidal one. An opposite result was observed by performing the same analysis for *S.aureus* Mu50.

Furthermore, the PB4, PB5, PB7, and PB8 compounds were tested on *S. aureus* ATCC12598 and *E. faecalis* VRE ATCC51299 at different *inocula* ranging from 1 × 10 to 1 × 10^7^ CFU/mL, considering that the standard inoculum for *S. aureus* and *Enterococcus* spp. was 1 × 10^5^ (Appendix A).

The tests performed revealed an inoculum effect (IE). The PB4 and PB5 compounds showed an IE on *S. aureus* ATCC12598 in a range between <0.015 and 4 mg/L and from 0.03 to 8 mg/L respectively, while the IE on *E. faecalis* VRE ATCC51299 was evident for inocula lower than 1 × 10^5^ in a range from 0.06 to 0.5 mg/L for PB4 and from 0.5 to 4 mg/L for PB5.

PB7 and PB8 exhibited an IE on *S. aureus* ATCC 12598 in a range from 2 to 32 mg/L and from 0.5 to 4 mg/L, which was most evident at the highest and lowest inoculum concentrations, 1 × 10^7^ and 1 × 10^1^ respectively.

PB7, tested on *E. faecalis* ATCC51299, only had an IE at inoculum concentration higher than 1 × 10^5^ in a range from 8 to 32 mg/L, while PB8 acted on *E. faecalis* ATCC51299 in a range from 0.5 to 32 mg/L with an IE at the highest and lowest inoculum concentrations, namely 1 × 10^7^ and 1 × 10^1^ respectively.

#### 2.3.2. Antimicrobial Activity of PB Compounds against Gram-Negative Strains

We included ATCC strains belonging to different species, namely *A. baumannii* ATCC17978; *E. coli* ATCC25922, *P. aeruginosa* ATCC27853; and two strains of *K. pneumoniae*: ATCC700603 and the MDR strain ATCC BAA-2814.

As showed in Table 2, compounds PB1, PB2, PB3, PB5, PB6, PB7, and PB8 had MIC values ranging from 32 to 128 mL/L. Only PB4 showed good effectiveness in contrasting bacterial growth in *E. coli* and *A. baumannii,* with an MIC value of 0.25 mg/L and a good bactericidal action, with an MBC value of 0.5 mg/L. 

The behavior of molecules was variable when tested on *E. coli* ATCC 25922: PB1 and PB3 acted at 16 mg/L, PB5 and PB7 at 32 mg/L, PB2, and PB8 at 128 mg/L. PB4 was the only molecule of those tested to have bacteriostatic action, with a MIC value of 1 mg/L, and t bactericidal action, with an MBC value of 4 mg/L.

For *P. aeruginosa*, all compounds showed MIC values comprised between 32 and 128 mg/L. For the two different strains of *K. pneumoniae*, MIC values were approximately 64 to 128 mg/L.

As previously described for Staphylococchi and Enterococchi, we tested the IE effect, but decided to focus on the PB4 compound as it showed the best antimicrobial performance on *E. coli* and *A. baumannii.*

All strains exhibited an IE for inoculum concentrations lower than 1 × 10^5^ CFU/mL only, in particular: *A. baumanni* ATCC17978 and *E. coli* ATCC25922 showed similar ranges, from <0.25 to 1 mg/L and from 0.5 to 2 mg/L respectively (Appendix A).

### 2.4. Evaluation of Cell Cytotoxicity by MTT Assay

To evaluate the biological activity of the PBn series, compounds were screened for toxicity against the colon-rectal cancer cell line CaCo-2 (ATCC HTB-37).

These human cancer cells were treated with solutions of the compounds at decreasing concentrations, starting from 100 μM up to 0.01 μM. The reference compound used for these assays was 5-Fluorouracil (5-Fu), a pyrimidine analogue belonging to the family of antimetabolite.

The antiproliferative activity of these solutions was evaluated after 24 h of incubation, performing 3-(4,5-dimethylthiazol-2-yl)-2,5-diphenyl tetrazolium bromide (MTT) assays to obtain cell growth curves and IC50 values, reported in Figure 2 and Appendix A, respectively.

The data showed that all compounds of the PBn series possess high ability to influence cell growth, as their IC50 values were significantly lower than the well-known 5-FU.

The PB2-4 compounds were highly cytotoxic, with IC50 values oscillating between 0.32 and 0.87 μM. PB4 was the compound with the most promising IC50, 0.32 uM, showing a cytostatic activity two order of magnitude lower than 5-FU.

The PB1 and PB7 compounds exhibit moderate toxicity, with IC50 values of 1.2 and 1.36 μM, respectively. Noteworthy, PB5, PB6, and PB8 were the less antiproliferative compounds, showing IC50 values of 3.90, 6.80, and 5 μM, respectively; nevertheless, their activity was significantly higher than 5-FU (27.36 μM).

## 3. Discussion and Conclusions

In the first phase of this work, attention was paid to the construction of a QSAR model, through the VolSurf+ software, for the evaluation of the microbiological activity of heteroaryl-ethylenes. Although few data have been reported for this kind of synthetic derivative [22,26,27,28,29,30,31,32,33,34,35], it was possible to take advantage of the data available for stilbenic molecules mainly derived from natural sources. One of the most investigated bacterial strain is *S. aureus* ATCC29213, and this allowed us to build up the PLS model; the aim of this kind of statistical and molecular modelling approach is to highlight promising scaffolds. The projection of an in-house database of heteroaryl-ethylenes on the new activity model allowed us both to select six structures with different chemical structure from an in-house database of etheroaril-hethylenes and to design two new molecules, which were synthesized, purified, and characterized (see Section 4 and Appendix A). 

Antibiotics resistance was one of the major concerns in the presence of bacterial infection, especially those caused by methicillin-resistant *S. aureus* and *Enterococcus* spp. resistant to vancomycin, teicoplanin, linezolid, and daptomycin, and those caused by *Enterobacterales*, *Acinetobacter baumannii,* and *Pseudomonas aeruginosa* resistant to carbapenems and polymyxins [12,13,42,43].

Our compounds showed a wide spectrum of action; in particular, PB2, PB3, and PB6 were active with different ranges of MIC on *S. aureus* and *Enterococcus* spp.; PB6 on *S. aureus* strains; PB5 and PB8 showed good MIC values (ranging from 0.5 to 4 mg/L for PB5 and 2 to 4 mg/L for PB8) for *S. aureus* and *Enterococcus* spp; PB1 instead was only active on *S. aureus* not resistant to methicillin. The most promising molecule was PB4, active on *S. aureus*, *Enterococcus* spp., *E. coli,* and *A. baumannii*, with the MIC values between 0.25 and 1 mg/L, the lowest among all of the molecules tested. These promising findings were also confirmed by MBC and by the results obtained testing for inoculum effect.

According to the definition of bactericidal/bacteriostatic proprieties (bacteriostatic activity has been defined as a MBC/MIC ratio >4, but several technical problems and other factors can affect determination of the ratio) [44], PB4 and P8 showed bactericidal effect (MBC/MIC ≤ 4), whereas PB5 and PB6 showed a strain-dependent bactericidal/bacteriostatic activity. 

The evaluation of the toxicity of the molecules of this set towards a human colorectal cancer (CaCo-2) cell line was comparable to the results obtained from a known anticancer drug, namely 5-Fluorouracil. For PB1 to BP6, the results were in line with data previously reported for human breast cancer cell line [18,19,20]. All molecules proved to be much more active than 5-Fluorouracil against CaCo-2 cells. Under the conditions tested, the less active molecule inhibits the growth and replication of cancer cells at a concentration corresponding to 1/5 of that required for 5 Fluorouracil.

After this preliminary evaluation of the biological activity of the designed compounds, our next steps in this study will certainly involve: (i) testing these molecules on clinical strains of MRSA, *Enterococcus* spp., *E. coli,* and *A. baumannii*; (ii) evaluating the inhibition of bacterial and cellular replication at the same time; (iii) understanding the mechanisms by which these compounds exert their antibacterial and antitumor activity to gain a deeper insight into the role of bacterial infections in the development of cancer. Particular attention will be devoted to the most promising compound, namely PB4, that showed promising results for the treatment of both Gram-positive and Gram-negative bacterial infections.

As evidenced by the molecular modelling studies, this compound has the right balance between charge state and hydrophobic/hydrophilic region descriptors and may lead to the development of new drugs. However, these structural features may enable oxidation/hydroxylation through bacterial multicomponent monooxygenases (BMMs), that catalyze oxidation/hydroxylation of hydrocarbon substrates in *Pseudomonas aeruginosa* and *Klebsiella pneumoniae* strains [45]. This explains the low antimicrobial activity observed towards these Gram-negative bacterial strains.

## 4. Materials and Methods

### 4.1. General

All solvents and reagents, 5,6,7,8-tetrahydroisoquinoline, and 5-(3-chlorophenyl) furan-2-carbaldehyde were obtained from Sigma Aldrich and were used as received. PB1-6, 2-ethyl-5,6,7,8-tetrahydroisoquinolin-2-ium iodide, and 5′-(dibuthylamino) [2,2′-bithiophene]-5-carbaldehyde were prepared according to literature procedures [18,19,20,46,47]. All reactions were carried out under nitrogen atmosphere unless otherwise stated. Thin-layer chromatography (TLC) was performed on Merck silica gel plates with F-254 indicator. Column chromatography was performed on Silica gel 60. ^1^H and ^13^CNMR spectra were recorded at 27 °C using a Varian Inova 500 spectrometer. Chemical shifts (δ) are expressed in ppm and referenced to residual undeuterated solvent. 

### 4.2. Microorganisms and Growth Conditions

Control strains *S. aureus* ATCC29213, *S. aureus* ATCC12598, *S. aureus* ATCC BAA-1556 (USA300), *S. aureus* ATCC 700699 (Mu50-NRS1), *E. faecalis* ATCC29212, *E. faecalis* ATCC51299, *A. baumannii* ATCC17978; *E. coli* ATCC25922, *P. aeruginosa* ATCC27853, *K. pneumoniae* ATCC700603; and the MDR strain *K. pneumoniae* ATCC BAA-2814 were obtained from the American Type Culture Collection (Manassas, VA, USA).

*S. aureus* was grown on Mannitol Salt Agar (CM0085B, Thermo Scientific™ Oxoid™, Basingstoke, UK); *E. faecalis* and *E. faecium* on Bile Esculine Agar (CM0888, Thermo Scientific™ Oxoid™, Basingstoke, UK); and *A. baumannii*, *E. coli*, *P. aeruginosa*, and *K. pneumoniae* on MecConkey agar (CM0007, Thermo Scientific™ Oxoid™, Basingstoke, UK) at 37 °C for 24 h.

### 4.3. Minimum Inhibitory Concentration (MIC) and Minimum Bactericidal Concentration (MBC)

Microtiter plate assays were performed to determine the minimum inhibitory concentration of the PB molecules according to the standard method [48], with some modifications. 

All compounds were at a concentration of 8.000 mg/L in 100% dimethylsulfoxide DMSO (85190, Thermo Scientific™ Oxoid™, Basingstoke, UK). Further dilutions of the substances were prepared using Mueller Hinton II Broth Cation-Adjusted (CA-MHB) (212322, BD BBLTM, Franklin Lakes, NJ, USA). The tested concentration range was between 128 mg/L and 0.25 mg/L. A 100-μL aliquot of each PB molecules was inoculated in 96-well microplates containing sterile CA-MHB and serial dilutions were performed.

Subsequently, starting from different 0.5 Mc Farland (10^8^ CFU/mL) bacterial suspension, scalar dilutions of the inoculum were carried out in order to evaluate the presence of an inoculum effect. The microplates were incubated for 18 ± 2 h at 37 °C. 

MIC values were determined at the lowest antimicrobial agent concentration inhibiting bacterial growth, MBCs at the lowest concentration of each antimicrobial agent resulting in microbial death, as defined by the inability to re-culture bacteria.

To perform MBC assays, the entire volume of the well containing no visible bacterial growth (MIC value), and three dilutions higher and lower than the MIC value, were plated in Tryptone Soya Agar (CM013, Thermo Scientific™ Oxoid™, Basingstoke, UK).

The MIC and MBC values were expressed in mg/L. The tests were repeated in duplicate for all substances, and two further MICs were performed on PB4, PB5, PB7, and PB8, as these were considered to be the most promising compounds of the entire series.

### 4.4. Evaluation of the Anticarcinogenic Activity of the PB Series on Human Colorectal Adenocarcinoma Cells

To evaluate the effect of the substances, an MTT ([3-(4,5-dimethylthiazol-2-yl)-2,5-diphenyltetrazolium bromide]) assay was performed as previously described (Biomolecules). Briefly, human colorectal adenocarcinoma cells (CaCo-2 HTB-37^TM^ Colorectal Adenocarcinoma Human, American Type Culture Collection, Manassas, VA, USA) were grown in Dulbecco’s MEM (DMEM) with 10% heat-inactivated fetal bovine serum, 2 mM L-Alanyl-L-Glutamine, penicillin-streptomycin (50 units-50 μg for mL) and incubated at 37 °C in a humidified atmosphere of 5% CO_2_, 95% air. CaCo-2 cells were plated in 96 well pates and incubated at 37 °C. The PBn series compounds and 5-FU were prepared as a 1 mM solution in 10 mL with 0.01% DMSO. Then, 24 h after plating, cells were treated with 20 μL of each solution. Untreated cells were used as controls. Microplates were incubated at 37 °C in a humidified atmosphere of 5% CO_2_, 95% air for 24 h, and then cytotoxicity was measured with colorimetric assay based on the use of tetrazolium salt MTT (3-(4,5-dimethylthiazol-2-yl)-2,5-diphenyl tetrazolium bromide. IC50: this parameter expresses the concentration of the tested compound necessary to kill half of the cell population after 24 h of incubation relative to untreated controls. The results were read on a multiwell scanning spectrophotometer (AF2200 plate reader, Eppendorf, Milan, Italy), using a wavelength of 569 nm. Each value was an average of 4 wells. The IC50 values were calculated by nonlinear regression analysis using the GraphPad Prism 6.0 software.

## Figures and Tables

**Figure 1 antibiotics-10-01034-f001:**
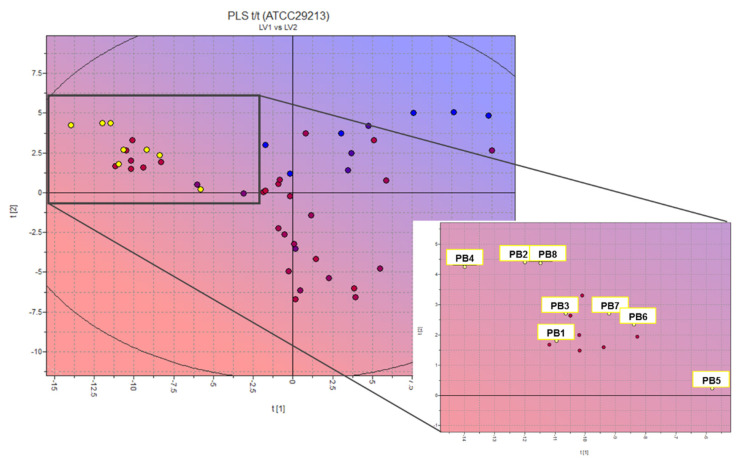
Two-dimensional scores plot of the PLS activity model at second latent variable and projection of designed compounds (inset: zoom on PBn projection). Color-scale by cytotoxic activity value: cyan/blue, low antimicrobial activity; pink/red, high antimicrobial activity; yellow circles: designed PBn compounds.

**Figure 2 antibiotics-10-01034-f002:**
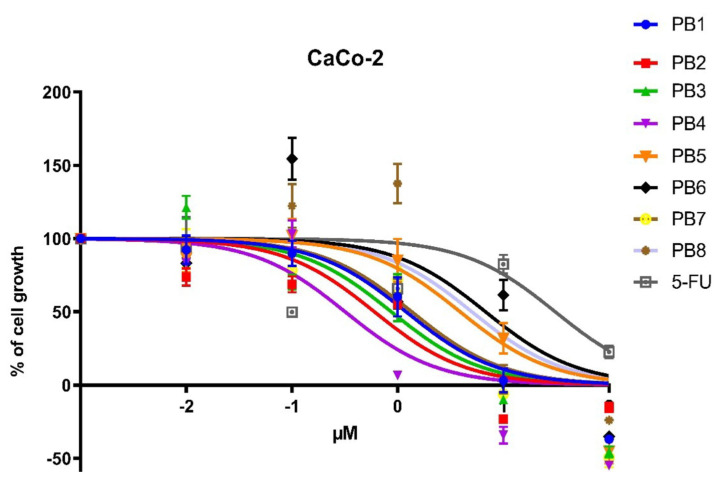
Dose–response curves of antiproliferative activities of PBn compounds and 5-FU against CaCo-2 cells after 24-h treatment.

**Table 1 antibiotics-10-01034-t001:** MIC and MBC values for Gram-positive control strains.

mg/L (μM)												
Strain	PB1	PB2	PB3	PB4		PB5		PB6	PB7		PB8	
	MIC	MIC	MIC	MIC	MBC	MIC	MBC	MIC	MIC	MBC	MIC	MBC
*S.aureus* Mu50-VISA	32 (76.87)	8 (13.59)	32 (70.28)	0.25 (0.49)	2 (3.91)	2 (3.70)	4 (7.40)	4 (8.73)	16 (34.50)	16 (34.50)	2 (3.46)	8 (13.83)
*S.aureus* USA300-CAMRSA	2 (4.80)	4 (6.80)	2 (4.39)	0.25 (0.49)	2 (3.91)	0,5 (0.93)	4 (7.40)	4 (8.73)	2 (4.31)	32 (69.00)	2 (3.46)	4 (6.91)
*S.aureus* ATCC29213-MSSA	4 (9.61)	4 (6.80)	4 (8.78)	0.25 (0.49)	2 (3.91)	1 (1.85)	4 (7.40)	4 (8.73)	4 (8.63)	4 (8.63)	2 (3.46)	8 (13.83)
*S.aureus* ATCC12598-MSSA	2 (4.80)	4 (6.80)	8 (17.57)	0.25 (0.49)	2 (3.91)	4 (7.40)	4 (7.40)	8 (17.46)	8 (17.25)	8 (17.25)	2 (3.46)	4 (6.91)
*E. faecalis* ATCC51299-VRE	32 (76.87)	16 (27.18)	32 (70.28)	0.5 (0.98)	4 (7.82)	4 (7.40)	128 (236.94)	32 (69.85)	8 (17.25)	128 (276.02)	4 (6.91)	4 (6.91)
*E. faecalis* ATCC29212-VSE	32 (76.87)	16 (27.18)	32 (70.28)	0.5 (0.98)	4 (7.82)	4 (7.40)	128 (236.94)	32 (69.85)	8 (17.25)	128 (276.02)	4 (6.91)	4 (6.91)

VISA: vancomycin intermediate *S. aureus*; CAMRSA: community acquired methicillin resistant *S. aureus;* MSSA: methicillin susceptible *S. aureus*; VRE: vancomycin resistant enterococci; VSE: vancomycin susceptible enterococci.

**Table 2 antibiotics-10-01034-t002:** MIC and MBC values for Gram-negative control strains.

mg/L (μM)									
Strain	PB1	PB2	PB3	PB4		PB5	PB6	PB7	PB8
	MIC	MIC	MIC	MIC	MBC	MIC	MIC	MIC	MIC
*A. baumannii* ATCC17978	128 (307.47)	32 (54.37)	64 (140.55)	0.25 (0.49)	0.5 (0.98)	>128 (>236.94)	64 (137.70)	32 (69.00)	128 (221.22)
*E. coli* ATCC25922	16 (38.43)	128 (217.47)	16 (35.14)	1 (1.96)	4 (7.82)	32 (59.24)	>128 (>279.40)	32 (69.00)	128 (221.22)
*P. aeruginosa* ATCC27853	>128 (>307.47)	32 (54.37)	>128 (281.10)	64 (125.14)	-	>128 (>236.94)	>128 (>279.40)	>128 (>276.02)	>128 (>221.22)
*K. pneumoniae* ATCC700603	>128 (>307.47)	>128 (>217.47)	>128 (>281.10)	>128 (>250.27)	-	>128 (>236.94)	>128 (>279.40)	128 (276.02)	>128 (>221.22)
*K. pneumoniae* ATCC BAA-2814	>128 (>307.47)	>128 (>217.47)	128 (>281.10)	64 (125.14)	-	>128 (>236.94)	>128 (>279.40)	64 (138.01)	>128 (>221.22)

## Data Availability

Not applicable.

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
