# Peer review of "Heteroaryl-Ethylenes as New Lead Compounds in the Fight against High Priority Bacterial Strains"

_antibiotics, 2021, doi:10.3390/antibiotics10091034_

Round 1
Reviewer 1 Report
- General comment
In the manuscript, antimicrobial activities (MIC and MBC) of eight heteroaromatic stilbene derivatives (PB1~8) on Gram-positive and Gram-negative ATCC strains were investigated, and PB4, PB5, PB7 and PB8 showed the best values in terms of MIC. In addition, these derivatives were active in reducing CaCo-2 cells viability. The results suggest that heteroaromatic stilbene derivatives are the promising candidates for further studies as a new class of antibiotic compounds.
2. Major revision
1) Line 117~146 and Figure 1
- Line 119: It is recommended to revise a reference [35] to [36].
- 1: As it is very difficult to understand the unit of axis, t[1] of transverse axis and t[2] of longitudinal axis, it is essential to explain more precisely and intelligibly at the legend of Fig. 1.
- Line 118~146: As it is very difficult to understand VolSurf+ Analysis, including PLS model, PLS t1/t2 scores plot, 3D pharmacophoric descriptors (DODODO, ACDODO and DRDODO) and H-bond donors volume (WO1 and WO2), it is essential to explain more precisely and intelligibly at the section of Materials and Methods, similarly to “4.1.1. VolSurf+ descriptors” in Ref.18).
2) Table 1, 2, S1, S3 and S4
The concentration of PBn compounds was shown as µM in Figure 2, and the molecular weight of PB1-8 was different each other. Therefore, it is essential to show MIC and MBC values as µM (mg/L) or µM.
- Minor revision
1) Line 106: Revise “C/oncentration” to “C/oncentration”
Author Response
Point by point responses to Reviewer’s comments
Reviewer 1
- General comment
In the manuscript, antimicrobial activities (MIC and MBC) of eight heteroaromatic stilbene derivatives (PB1~8) on Gram-positive and Gram-negative ATCC strains were investigated, and PB4, PB5, PB7 and PB8 showed the best values in terms of MIC. In addition, these derivatives were active in reducing CaCo-2 cells viability. The results suggest that heteroaromatic stilbene derivatives are the promising candidates for further studies as a new class of antibiotic compounds.
We thank the reviewer for the comments.
2. Major revision
1) Line 117~146 and Figure 1
- Line 119: It is recommended to revise a reference [35] to [36].
All the references have been checked and updated. - 1: As it is very difficult to understand the unit of axis, t[1] of transverse axis and t[2] of longitudinal axis, it is essential to explain more precisely and intelligibly at the legend of Fig. 1.
The basic concept of VolSurf+ is to compress the information present in the molecule 3D maps into a total of 128 numerical descriptors related to the physicochemical property.
The plot reported in Figure 1 represent the molecules of both the training and the test sets in the two-dimensional space of the Latent Variables (LVs) t[1] and t[2], which are linear combinations of the original physicochemical property descriptors. For these reasons, the LVs, t[1] and t[2], have no dimension and, therefore, no unit has been reported. - Line 118~146: As it is very difficult to understand VolSurf+ Analysis, including PLS model, PLS t1/t2 scores plot, 3D pharmacophoric descriptors (DODODO, ACDODO and DRDODO) and H-bond donors volume (WO1 and WO2), it is essential to explain more precisely and intelligibly at the section of Materials and Methods, similarly to “4.1.1. VolSurf+ descriptors” in Ref.18).
The Partial Least Squares (PLS) is a chemometric tools for extracting and rationalizing the information from any multivariate description of a biological system. The description of this chemometric procedure is out of the scope of this section. However, to clarify some of key points, we welcome the reviewer’s suggestions and we have inserted a new paragraph in the Supplementary materials.
2) Table 1, 2, S1, S3 and S4
The concentration of PBn compounds was shown as µM in Figure 2, and the molecular weight of PB1-8 was different each other. Therefore, it is essential to show MIC and MBC values as µM (mg/L) or µM.
The Tables have been revised: the MIC and MBC concentration values have been reported in both units: mg/L and µM, in square brackets.
- Minor revision
1) Line 106: Revise “C/oncentration” to “C/oncentration”
This typo has been corrected.

Reviewer 2 Report
The paper falls in the Journal's frame and could be of interest for the readers!
Introduction: well presented and linked with the topic. Please add a phrase about previous identified heteroatomic stilbene derivatives by the collective.
M & M: well presented
Discussions: In general well presented as antimicrobial activity of PBn's but please insist more on the cytotoxicity and potential deleterious activity of these compounds.
References: In general associated to the topic.
Author Response
Reviewer 2
The paper falls in the Journal's frame and could be of interest for the readers!
Introduction: well presented and linked with the topic. Please add a phrase about previous identified heteroatomic stilbene derivatives by the collective.
M & M: well presented
Discussions: In general well presented as antimicrobial activity of PBn's but please insist more on the cytotoxicity and potential deleterious activity of these compounds.
References: In general associated to the topic.
We thank the reviewer for the comments.
